# Carbon Emission Evaluation of Roadway Construction at Contaminated Sites Based on Life Cycle Assessment Method

Luorui Zheng [1], Yingzhen Li [2], Cheng Qian [1] and Yanjun Du [2,*]

1   School of Transportation, Southeast University, Nanjing 211189, China; 213201480@seu.edu.cn (L.Z.);
    213201453@seu.edu.cn (C.Q.)
2   Jiangsu Key Laboratory of Urban Underground Engineering and Environmental Safety,
    Institute of Geotechnical Engineering, Southeast University, Nanjing 211189, China; liyingzhen@seu.edu.cn
*   Correspondence: duyanjun@seu.edu.cn

**Abstract:** Greenhouse gas emissions induced by climate change have garnered global attention. Minimizing climate change can be achieved through the reduction of carbon emissions in transportation infrastructure construction and in the production of construction materials. This study aims to calculate carbon emissions in three hypothetical construction scenarios based on the life cycle assessment (LCA) method when a roadway passes across polluted soil at contaminated sites. Three methods are employed to remediate contaminated soil: off-site cement kiln co-processing, on-site ex-situ thermal desorption, and on-site ex-situ solidification/stabilization. Carbon emissions are calculated using the LCA method for each scenario. The baseline carbon emission is estimated for the scenario in which contaminated soil is remediated using the off-site cement kiln co-processing method, and the roadway subgrade is constructed using transported clean soil. In the other two scenarios, contaminated soils are remediated using the on-site ex-situ thermal desorption and solidification/stabilization methods, respectively, and then they are reused as roadway subgrade materials. The LCA analyses demonstrate that the total carbon emission reductions range from 1168.48 to 2379.62 tons per basic unit, corresponding to decreased of 19.31% to 39.33%, respectively, compared to baseline. The reuse of solid waste to replace sand and ordinary Portland cement (OPC) as raw materials in roadway construction reduces carbon emissions by 498.98 tons. Finally, a comparison of carbon emissions between the three scenarios indicates that reducing carbon emissions in the remediation of contaminated soil and reusing solid waste as construction materials are two important methods for achieving overall carbon emission reductions in roadway construction projects.

**Keywords:** life cycle assessment; soil remediation; solid waste; roadway; carbon emission

## 1. Introduction

Global warming poses a significant challenge to humanity, resulting in adverse effects on the climate, rising sea levels, and both natural and human environments. In 2015, the signing of the Paris Agreement [1] led most countries to commit to limiting greenhouse gas (GHG) emissions and developing strategies for their reduction. China has set ambitious goals to reach peak carbon emissions by 2030 and achieve carbon neutrality by 2060 [2]. The transportation sector, following industry, has become the second-largest contributor to carbon emissions [3]. Within transportation, the construction phase of roadway projects significantly contributes to overall GHG emissions [4]. Roadway construction has been identified as a major source of GHG emissions in Europe and America, primarily due to resource-intensive practices [5]. Similarly, in developing countries like China, high-energy materials, extensive material processing, and a large fleet of vehicles and equipment used for transportation and on-site construction activities contribute to substantial carbon emissions during roadway construction [6]. Consequently, reducing carbon emissions during roadway construction is of the utmost importance for nations committed to mitigating GHG emissions.

Due to China's urban expansion and industrial restructuring, the relocation of industrial enterprises has resulted in the abandonment of heavily contaminated sites [7]. This leads to significant wastage of soil resources and environmental pollution, thereby impeding the efficient circulation and utilization of land resources. In China, cement kiln co-processing is a commonly employed method for treating contaminated soil [8] and solid waste due to its simplicity and low cost. Thermal desorption technology is extensively used in remediation of organically contaminated soil projects and offers notable advantages. Solidification/stabilization (S/S) is a widely adopted technique for remediating heavy metal-contaminated soil, known for its convenience, cost effectiveness, and quick results [9]. Cement-based S/S technology has been proven effective in immobilizing heavy metals, even without additional additives [10]. The use of alternative additives such as fly ash, lime, and volcanic ash materials instead of cement [11] can not only enhance strength and reduce leaching, but it also significantly contributes to reducing carbon emissions during production. Following thermal treatment and stabilization/solidification, the performance of contaminated dredged soil meets the required acceptance criteria for its application as a construction material in various pavement layers [12]. Experimental evidence demonstrates that oil-contaminated soil treated with S/S can be utilized as a sub-base material for road pavement [13].

Currently, the application of solid waste in buildings and infrastructure is worth encouraging and further promoting [14]. China is facing a significant challenge regarding the production and disposal of solid waste. Ordinary Portland cement (OPC) is the most commonly used cementing agent in roadway construction. The production of OPC is associated with substantial GHG emissions, primarily due to clinker manufacturing. However, solid waste holds the potential to be a valuable resource for the development of green materials [15]. Solid waste such as recycled aggregates can be effectively utilized in the construction of roadway base layers and as a filling material for geosynthetic reinforced structures [16,17]. For instance, calcium carbide residues can be employed to improve wet clay and meet the performance requirements of roadbed fill soil [18]. Waste materials like steel slag and fly ash can also serve as construction materials for roadway pavement, achieving carbon emission reductions while enhancing roadway performance. Moreover, some coarse and fine aggregates in cement can be replaced with alternative materials like fly ash and blast furnace slag [19]. This approach helps reduce the demand for raw materials, conserves resources, and minimizes waste generation [20].

Life cycle assessment (LCA) provides an excellent method for understanding the sustainability issues of infrastructure systems and offers quantitative and transparent results to facilitate informed decision-making by designers. Previous studies have utilized LCA to design and select pavement types [21]. As environmental concerns become increasingly urgent worldwide, more researchers are using LCA to evaluate the environmental impact of roadway infrastructure construction, particularly in relation to carbon emissions [22]. For instance, Liu et al. analyzed the carbon emissions of roads with different structures and compared their differences [23]. In a study on carbon emissions during urban roadway construction, Mao et al. found that material use accounted for over 50% of the carbon emissions [24]. Research by Keijzer et al. on roadway infrastructure in the Netherlands revealed the significant potential for reducing carbon emissions in this field [25]. However, there is a relative scarcity of studies focusing on the application of LCA in calculating carbon emissions after the remediation and reuse of contaminated soil, with most studies concentrating on the methods used to remediate contaminated soil. This study aims to fill this gap.

Currently, most research focuses on the methods of remediating contaminated soil and the application of solid waste. However, there is a lack of comparative analysis regarding carbon emissions. The Chinese highway network has undergone continuous improvements. When a construction route passes across a contaminated site and its surroundings, redesigning the route would lead to unnecessary manpower and economic losses. In such cases, adopting the method of reusing remediated contaminated soil

becomes a preferable solution. Existing research does not provide clear insight into the design analysis of carbon emissions under the aforementioned engineering background.

The flowchart of this study, presented in Figure 1, illustrates the estimation of carbon emissions throughout the entire roadway project, encompassing material production, contaminated soil remediation, and roadway construction. This study utilizes LCA to analyze the total carbon emissions during roadway construction, as well as emissions at each stage. The study also discusses different emission results obtained and their influencing factors. The flowchart in Figure 1 provides a visual representation of the carbon emission estimation in the life cycle of roadway construction projects, which includes material production, contaminated soil remediation, and roadway construction. The findings of this study are valuable for informing strategies aimed at reducing carbon emissions in transportation infrastructure construction.

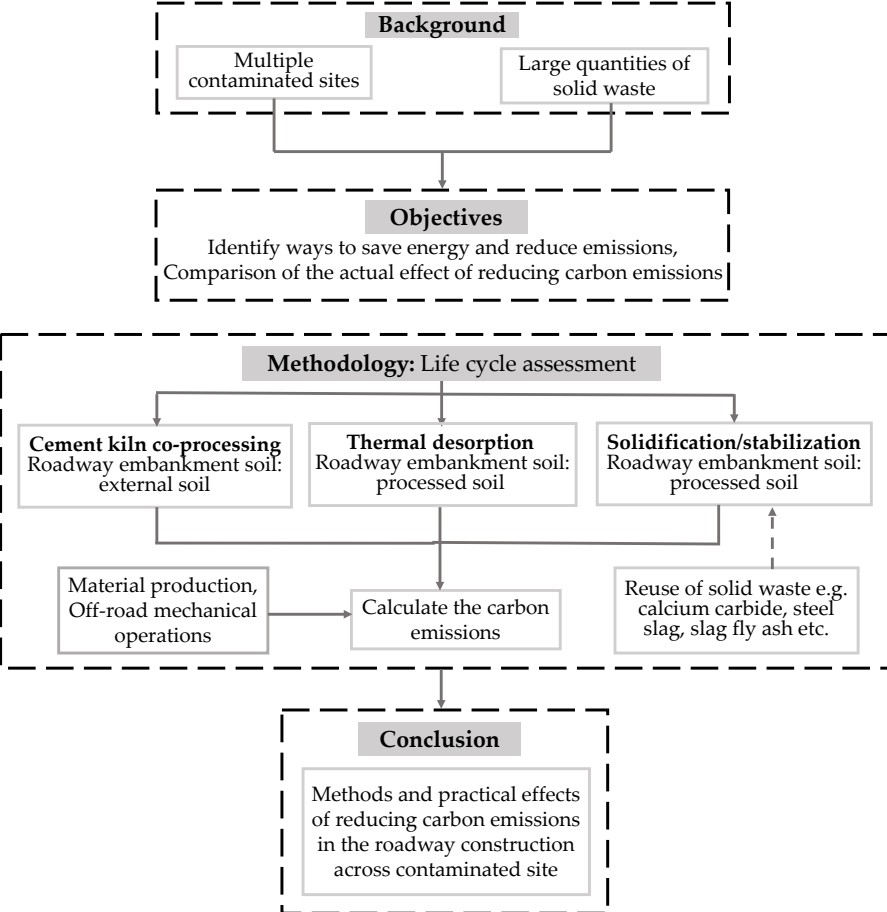

**Figure 1.** Flowchart of this study.

## 2. LCA Method

LCA is an analytical method comprising four stages: goal and scope definition, inventory analysis, impact assessment, and interpretation [26,27]. It is used to assess the inputs, outputs, and potential environmental impacts of a project throughout its entire life cycle [27]. Research on LCA primarily emerged in the mid-1980s [28], and the International Organization for Standardization [26,27] published the ISO 14040 series in the late 1990s, officially establishing LCA as a tool for measuring carbon emissions.

Goal and scope definition aims to clearly define the scope and purpose of the LCA study, as well as determine the environmental aspects to be evaluated. The research scale can be determined based on various factors, such as research conditions, depth, assumptions, system boundaries, constraints, and basic units. Inventory analysis involves quantifying the fundamental data related to the entire life cycle stage of the product,

including resource utilization, energy consumption, and emissions to the environment (such as waste gases, wastewater, solid waste, and other environmental releases). Impact assessment seeks to quantify different burdens in a standardized form or dimension, based on the results obtained from the inventory analysis stage. It estimates the energy and resources used in the system and evaluates the potential environmental impacts associated with these inputs and outputs. Interpretation primarily combines the issues identified during the inventory analysis and environmental evaluation with the research objectives. It draws final conclusions and provides recommendations based on these findings [29].

The LCA method for inventory analysis can be categorized into three main types: process-based LCA, input–output-based LCA, and hybrid LCA [30]. Process-based LCA relies on process analysis, which divides the system into a series of units and establishes corresponding data inventories to quantify the resource inputs and GHG emissions of each process. This method is widely used due to its convenience in independent analysis and has been standardized by the International Organization for Standardization. However, challenges remain in terms of complex data collection, defining system boundaries, and theoretical completeness. Nonetheless, process analysis still fulfills the evaluation requirements for individual products.

Researchers have proposed an input–output-based LCA method, called EIO-LCA [31], which links input–output tables from the US Department of Commerce with environmental data (including conventional pollutants, energy consumption, stone consumption, global warming potential, and ozone depletion potential). EIO-LCA utilizes the entire US economic system as the analytical system boundaries, considering the interactions between different industry sectors in the system. It can evaluate all direct and indirect economic impacts resulting from energy consumption and environmental emissions during product production. However, it is not suitable for analyzing the life cycle of individual products or activities.

To integrate the advantages of the aforementioned methods, Hendrickson et al. proposed a hybrid LCA method [32]. This approach converts process data into input–output analysis data for quantitative analysis, aiming to minimize accumulated errors and truncation errors that may arise from using the two methods independently. However, this method currently lacks effective database support and relies on input–output tables, which limits its application to single-product systems and makes it more suitable for complex product systems.

In comparison to the other two methods, process-based inventory analysis enables a more comprehensive and accurate calculation of carbon emissions throughout the life cycle of roadway construction passing across contaminated land. It encompasses the remediation of contaminated soil, material production, and construction processes. Previous studies on calculating carbon emissions in roadway construction have predominantly utilized process-based inventory analysis, affirming the reliability of this method. Therefore, in this study, we primarily employ process-based LCA to discuss and analyze the carbon emissions of highways, with the aim of reducing carbon emissions in the roadway construction industry [33].

## 3. Case Study

### 3.1. Background

This study presents several different scenarios for roadway construction involving the remediation of contaminated soil, material production, and engineering construction in three stages. Three scenarios are assumed based on the methods used for the remediation of contaminated soil. In Scenario 1, the traditional method was employed, where contaminated soil was transported to a cement kiln for incineration, and external soil materials were used as replacements. It was assumed that fully loaded soil trucks were transported for a distance of 10 km, while empty trucks were transported for 5 km. The carbon emission factor for processing contaminated soil in a cement kiln was selected as 350 kg/m$^3$, based on Xue et al.'s comprehensive research [34]. In Scenario 2, on-site ex-situ thermal desorption

technology was utilized to excavate contaminated soil and remediate organic-contaminated soil in a nearby temporary workshop [35]. The excavated soil was remediated and reused after passing an acceptance test. In Scenario 3, on-site ex-situ stabilization/solidification technology was employed to excavate and stabilize heavy metal-contaminated soil in a temporary workshop before reuse [36]. The soil transportation distance in Scenarios 2 and 3 was assumed to be 1 km. To effectively reduce carbon emissions, the study suggested reducing the use of high energy-intensive raw materials and reusing solid waste by incorporating them as raw materials for concrete production or roadbed filling. Fly ash was added to the concrete in significant amounts, with an optimal mix ratio of 68% based on experiments conducted by Irem Sanal [37]. Gu et al. found that reconstructing and activating secondary steel slag can enhance the performance of subgrade treatment. In their design study, a mixture consisting of 50% appropriately modified secondary steel slag, 45% soil, and 5% lime exhibited optimal performance [38].

This study was based on the scenario of a highway design route passing across a contaminated site. The concept was to either reuse remediated contaminated soil or transport clean soil from another site. The entire highway adopted an asphalt concrete pavement designed as a flexible pavement, and the schematic representation of a typical pavement structure is presented in Figure 2. The design speed was 80 km/h, with a lane width of 3.75 m and a shoulder width of 2.5 m. The half-width of the four-lane highway was 10 m. The road structure and performance adhered to national standards in China [39]. It was assumed that the average hauling distance from the asphalt mixing plant and materials yard to the construction site was 10 km, while the average hauling distance from the water-stable base mixing plant to the construction site was approximately 8 km.

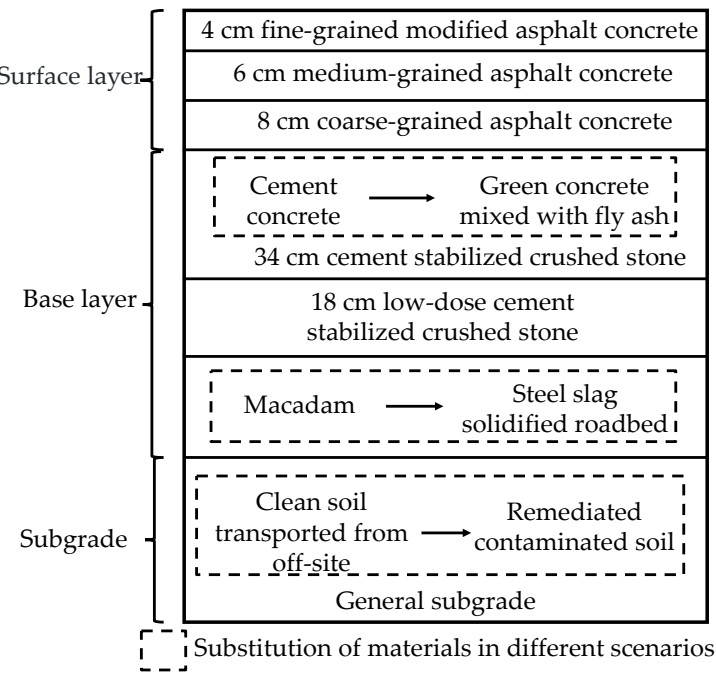

**Figure 2.** Schematic representation of typical pavement structure.

### 3.2. System Boundaries

This study employed the LCA method based on ISO 14040 [22,23] to quantify carbon dioxide emissions. Previous research did not include roadway use and end-of-life stages (EOL phase) [40,41], and data collection challenges have made long-term studies on reusing contaminated soil and solid waste for roadway construction rare. Additionally, representing infrastructure components such as bridges, tunnels, and traffic facilities using a unified basic unit is difficult. Therefore, this study excluded maintenance, operation, and end-of-life stages from the system boundaries, and carbon emission calculations for bridges,

tunnels, and ancillary facilities were not considered. The focus of this study was solely on carbon emissions during the life cycle of roadway construction, specifically pavement and subgrade engineering.

The total carbon dioxide emissions were calculated considering the project's life cycle system boundaries. In this study, the system boundaries mainly encompassed contaminated soil excavation and remediation, material production, and roadway construction stages. These stages were divided into three scenarios based on the different methods used for the remediation of contaminated soil, as depicted in Figure 3.

**Scenario 1 Cement kiln co-processing**

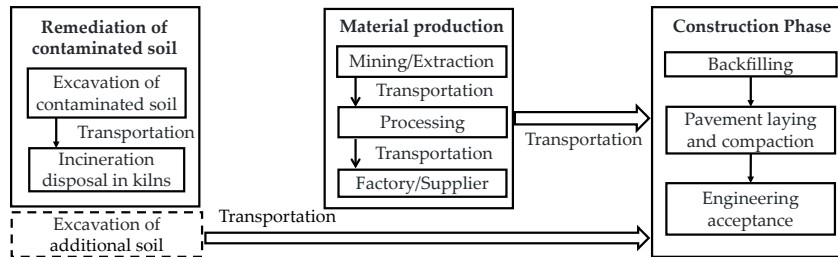

(**a**) Contaminated soil remediation using cement kiln co-processing.

**Scenario 2 Ex-situ thermal desorption**

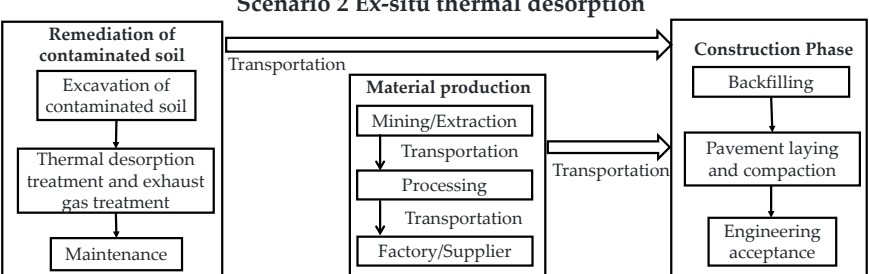

(**b**) Contaminated soil remediation using ex-situ thermal desorption.

**Scenario 3 Solidification/stabilization**

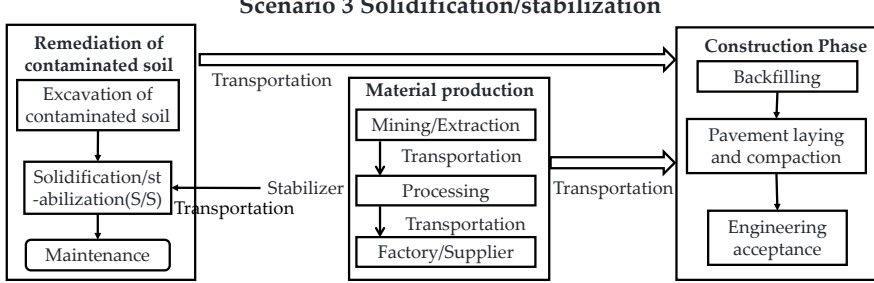

(**c**) Contaminated soil remediation using stabilization/solidification.

**Figure 3.** System boundaries of the life cycle of road infrastructure construction.

To represent the carbon emission results, this study considered the half-width of one kilometer of a two-way four-lane highway as one basic unit. The final presentation of the carbon emission results quantified the $CO_2$ emissions of one basic unit of the highway during the construction stage, expressed as t/basic unit.

### 3.3. Inventory Analysis and $CO_2$ Emissions Calculation

Based on the system boundaries and inventory analysis of carbon emissions during the life cycle of roadway construction after remediation of contaminated soil, the calculation of carbon emissions for stabilized contaminated soil was performed using Equation (1):

$$E_{\text{total}} = E_{\text{n}} + E_M + E_{\text{p}} \tag{1}$$

Here $E_{\text{total}}$ is the total carbon emissions (t) during roadway construction across a contaminated site, and $E_M$, $E_{\text{n}}$, and $E_{\text{p}}$ represent the carbon emissions (t) in the three stages of the roadway construction life cycle. To uniformly quantify and compare the different design scenarios, a basic unit carbon emissions value was used for representation, which was calculated using Equation (2):

$$E_{\text{fu}} = \frac{E_{\text{total}}}{l} \tag{2}$$

where $l$ is a conversion factor to ensure that the final result $E_{\text{fu}}$ is expressed in units of t/basic unit.

Special materials such as quicklime, activated carbon, NaOH, and others are necessary for the remediation of contaminated soil. The carbon emission factors for commonly used materials in soil remediation, based on the "IPCC National GHG Inventory Guidelines (2006)" [36], are provided in Table 1. Additionally, carbon emissions from off-road machinery need to be considered during the process. Soil material excavation, remediation of contaminated soil, and transportation are the primary sources of carbon emissions during the remediation process, with specific contributions varying depending on the remediation method employed. The calculation for total carbon emissions generated during contaminated soil remediation is given by Equation (3):

$$E_{\text{n}} = f_{\text{r}} \times V \tag{3}$$

where $E_{\text{n}}$ is the total carbon emissions (t) generated during the remediation of contaminated soil, r represents the remediation of contaminated soil method, $f_{\text{r}}$ is the carbon emission factor value (t/m$^3$) for the remediation method r, and $V$ is the volume of contaminated soil remediated (m$^3$).

This study primarily focused on the primary materials and high energy-consumption materials used in pavement construction, including cement, asphalt, sand, stone, and others. The carbon emissions from material production encompass the total carbon emissions generated during all upstream processes and activities preceding the utilization and processing of these materials [42]. The carbon emission factors for different types of materials are presented in Table 2. The calculation of the total carbon emissions accumulated from various materials is given by Equation (4):

$$E_M = \sum\nolimits_{\text{i}} (f_{\text{i}} \times M_{\text{i}}) \tag{4}$$

where $E_M$ is the total carbon emissions (t) from material production, i represents the type of material, $f_{\text{i}}$ is the carbon emission factor value (t/t) for material i, and $M_{\text{i}}$ is the amount (t) of material i used.

The carbon emissions generated by construction machinery and transportation vehicles are determined by measuring their actual fuel consumption. The hourly fuel consumption rate for construction machinery and transportation vehicles is obtained from publicly available highway project estimates compiled by China's national management agencies since 1958, representing average production conditions nationwide [23]. Specific values can be found in Table 3. Equation (5) is utilized to calculate the carbon emissions of off-road machinery:

$$E_{\text{p}} = \sum\nolimits_{\text{c}} (f_{\text{c}} \times u_{\text{c}} \times t) \tag{5}$$

where $E_{\text{p}}$ is the carbon emissions (t) generated by construction, c is the type of construction equipment, $f_{\text{c}}$ is the carbon emission factor value (t/kg) for the fuel used by equipment c, $u_{\text{c}}$ is the hourly fuel consumption rate (kg/h) for equipment c, and $t$ is the total working time (h) for equipment c.

The use of accurate and reliable life cycle emission factors is crucial for ensuring the accuracy and credibility of the results in studies. In this study, carbon emission factors were derived from the peer-reviewed literature and authoritative research reports. The selected emission factors were chosen to be universally applicable across different environ-

ments. They undergo comprehensive analysis processes and precise calculations, which contributed to the reliability of the relevant parameters and enhanced the credibility of the calculation results.

**Table 1.** Emission factors of materials involved in Scenarios 2 and 3 [43].

| Material/Fuel | Unit | Emission Factor |
|---|---|---|
| Water | kg/t | 0.17 |
| Quick lime | kg/kg | 1.34 |
| C30 concrete | kg/m$^3$ | 321.30 |
| Activated carbon | kg/kg | 9.97 |
| Naoh | kg/kg | 1.97 |
| Waste water | kg/t | 1.06 |
| MgO | kg/kg | 1.06 |
| Ca(OH)$_2$ | kg/kg | 1.017 |
| FeCl$_3$ | kg/kg | 0.18 |
| CaO | kg/kg | 0.50 |
| Al$_2$(SO$_4$)$_3$ | kg/kg | 0.50 |
| Al$_2$O$_3$ | kg/kg | 1.23 |
| Dicalcium phosphate | kg/kg | 2.70 |
| Fly ash | kg/kg | 0 |

**Table 2.** Emission factors of materials and fuels in roadway construction [23]. Data source: *a.* IPCC. 2006 [43]. *b.* Gong and Zhang, 2004 [44]. *c.* Zhang, 2002 [45]. *d.* Blomberg et al., 2011 [46]. Reproduced with permission from Yuanqing Wang, Journal of Cleaner Production; published by Elsevier Ltd., 2017.

| Material/Fuel | Unit | Emission Factor |
|---|---|---|
| Cement [a] | | |
| P.s. 32.5 | kg/t | 677.68 |
| P.o. 42.5 | kg/t | 920.028 |
| P.i. 52.5 | kg/t | 1041.577 |
| Building materials [b] | | |
| Sand | kg/m$^3$ | 2.56 |
| Rubble (cleft stone, block stone) | kg/m$^3$ | 3.37 |
| Gravel (2 cm, 4 cm, 6 cm, 8 cm) | kg/m$^3$ | 3.3 |
| Asphalt [c] | | |
| Bitumen | kg/t | 174.24 |
| Modified asphalt | kg/t | 295.91 |
| Fuel [d] | | |
| Diesel | kg/kg | 4.62 |
| Fuel oil | kg/kg | 4.36 |
| Electricity | kg/kwh | 0.983 |

**Table 3.** Characteristics of transportation and off-road machinery [23]. Reproduced with permission from Yuanqing Wang, Journal of Cleaner Production; published by Elsevier Ltd., 2017.

| Transportation Vehicle | Specification | Fuel/Energy Type | Fuel Efficiency | Unit (Per Vehicle Per 8 h) |
|---|---|---|---|---|
| Dump truck | 10–12 t | diesel | 58.46 | kg |
| Lorry | 10–12 t | diesel | 42.1 | kg |
| Off-road machinery | | | | |
| Bulldozer | 75–90 kw | diesel | 294.83 | kg |
| Excavator | 0.8–1.0 m$^3$ | diesel | 226.405 | kg |
| Loader | 1–1.5 m$^3$ | diesel | 280.28 | kg |
| Grader | 120–150 kw | diesel | 441.98 | kg |
| Road roller | 10–12 t | diesel | 165.18 | kg |
| Asphalt mixing station | 60–120 L | fuel oil | 2692.8 | kg |
| | | electricity | 1588.67 | kwh |
| Concrete batching plant | 50–60 L | electricity | 619.46 | kwh |

## 4. Results and Discussion

The evaluation results presented in Figure 4 illustrate the carbon emissions associated with the three different roadway scenarios. Among these scenarios, Scenario 1, which involved cement kiln co-disposal of contaminated soil, generated the highest carbon emissions, followed by on-site ex-situ thermal desorption remediation. The lowest carbon emissions among the three methods for remediating contaminated soil were observed using on-site ex-situ S/S. For each basic unit of construction, Scenario 1 produced total carbon emissions of 6050.774 t, Scenario 2 emitted 4882.290 t, and Scenario 3 had the lowest emissions at 3671.154 t. These findings highlight the significant impact of the chosen contaminated soil remediation method on carbon emissions in roadway construction. Scenario 1 showed total carbon emissions that were 1168.483 t higher than those of Scenario 2 and 2379.619 t higher than those of Scenario 3, representing reductions of 19.32% and 39.33%, respectively. Different methods of remediating contaminated soil resulted in notable variations in total carbon emissions. It is important to note that material production plays a prominent role in carbon emissions, which progressively increase as the proportion of carbon emissions from the remediation of contaminated soil decreases. Material production contributed to over 45% of the overall carbon emissions and could even account for up to 80% in Scenario 3. Due to limitations in construction efficiency and technology, the total carbon emissions from off-road machinery and transportation vehicles for each basic unit showed minimal differences across the scenarios. Carbon emissions during construction remained relatively consistent and did not exceed 10% of the total carbon emissions.

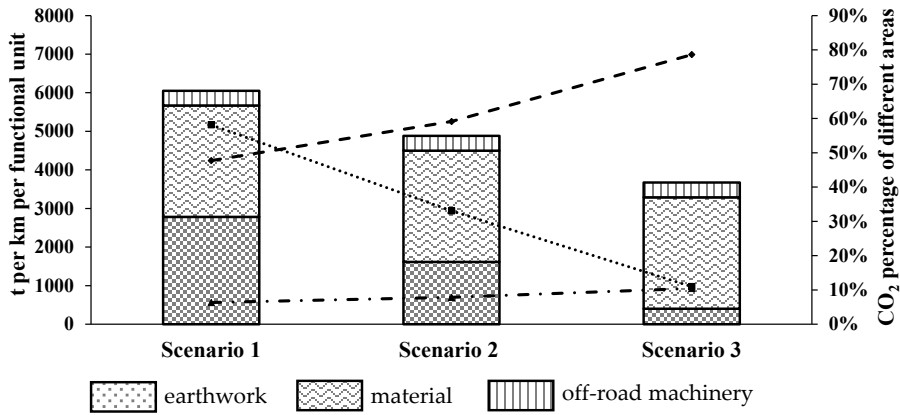

**Figure 4.** Main sources of $CO_2$ emissions in three scenarios.

In roadway engineering projects involving contaminated soil, controlling carbon emissions is crucial, especially during earthwork activities. Figure 5 illustrates the proportion of carbon emissions originating from the different components of contaminated soil and provides specific data on carbon emissions associated with earthwork activities. In Scenario 1, both the remediation of contaminated soil and material production contributed approximately 47% each to the carbon emissions. Notably, the cement kiln co-disposal method for contaminated soil was responsible for 43.96% of the total carbon emissions, amounting to 2782.166 tons. Cement kilns are commonly used in China for the disposal of various solid wastes and contaminated soil. Furthermore, the transportation of excavated contaminated soil to the cement plant and filled soil to the construction site accounted for 2.02% of the overall carbon emissions. Given the challenges in improving the cement kiln process without compromising remediation effectiveness, opting for a lower carbon remediation method would promote sustainable development. In Scenario 2, the proportion of carbon emissions from the remediation of contaminated soil decreased to 33.05%, with exhaust gas treatment accounting for a higher proportion of earthwork carbon emissions at 17.85%. The carbon emissions produced during the contaminated soil remediation process followed closely behind. Other accounted for only 0.33% of carbon emissions in Scenario 2 and could

be disregarded. Optimizing thermal desorption construction methods and improving the quantity and type of materials can effectively reduce overall carbon emissions.

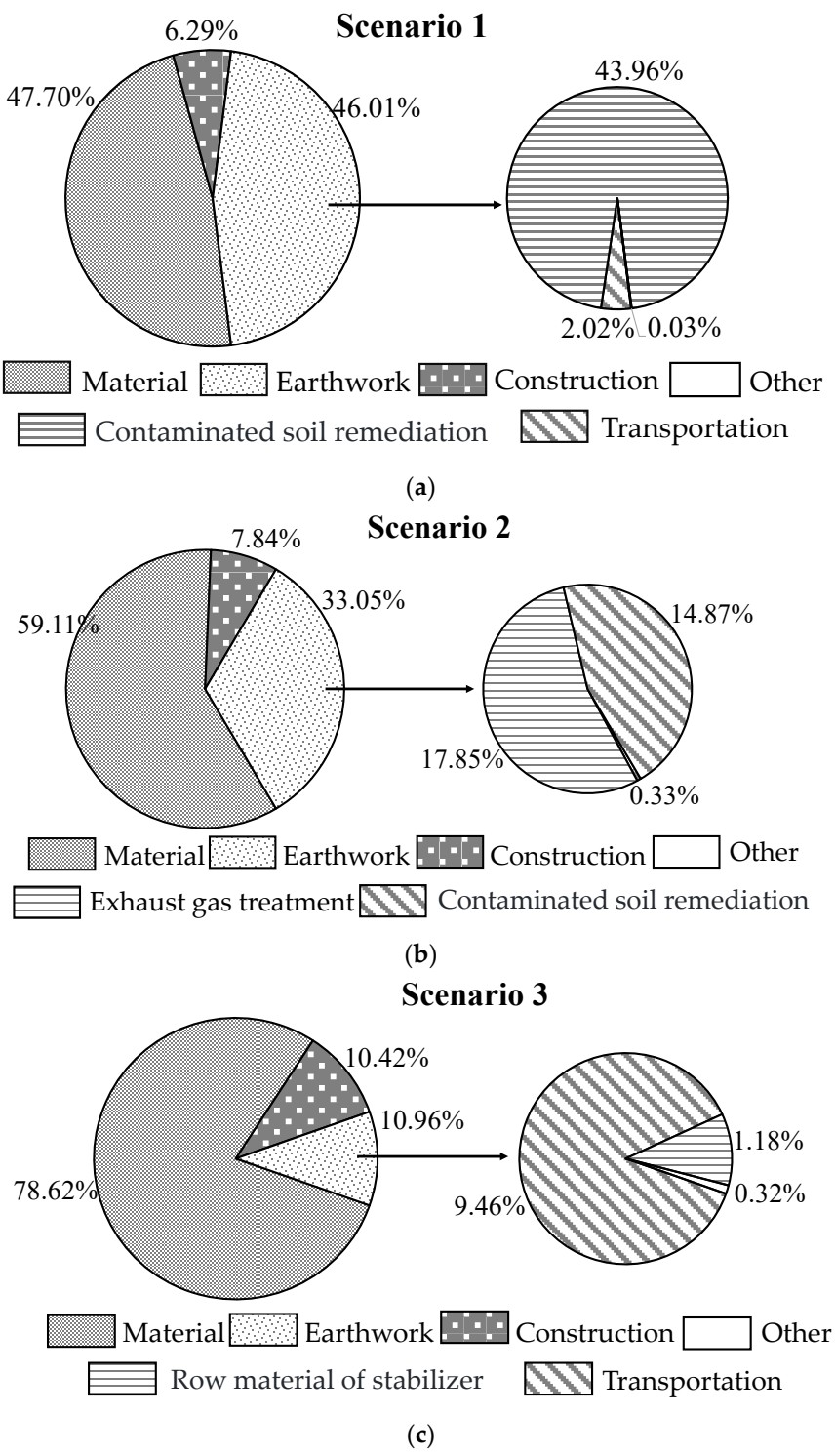

**Figure 5.** Components and proportions of carbon emissions in three scenarios: (**a**) cement kiln co-processing; (**b**) ex-situ thermal desorption; and (**c**) stabilization/solidification.

In Scenario 3, earthwork activities accounted for 10.96% of the overall carbon emissions, further reducing its proportion and resulting in the lowest emissions among the three scenarios. The production of raw materials for stabilizers was the primary contributor to carbon emissions from earthwork activities, with transportation of various stabilizer

raw materials contributing up to 1.18%. The proportion of carbon emissions from project implementation increased to 10.42%. Solid waste can be utilized as a raw material for S/S agents due to its high specific surface area, adsorption capacity, and ion exchange capabilities. For example, when combined with other materials, fly ash demonstrates excellent effects on heavy metal-contaminated soil [47]. To reduce carbon emissions in the S/S remediation of contaminated soil, it is crucial to employ low-carbon raw materials and enhance material processing technology.

Apart from earthwork engineering, the proportion of carbon emissions from material production gradually increased in all three scenarios. Reusing solid waste can effectively reduce carbon emissions in roadway construction. Figure 6 illustrates the proportion of $CO_2$ emissions from materials before and after implementing solid waste reuse. By reusing solid waste, the total carbon emissions associated with material production were reduced by 498.98 t, representing a decrease of 17.29%. In terms of asphalt concrete pavement materials, the primary source of carbon emissions was attributed to asphalt production.

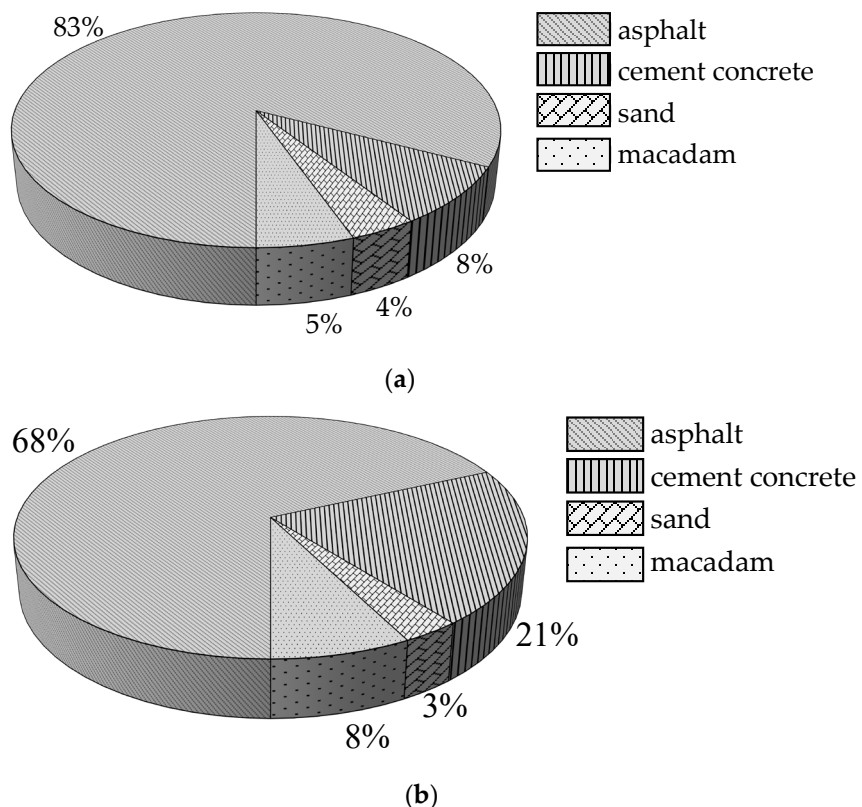

**Figure 6.** Proportion of $CO_2$ emissions from S/S material in (**a**) conventional design and (**b**) introducing solid waste under Scenario 3.

By reusing solid waste as a raw material in cement concrete production and as a filler in roadway structures, the proportion of carbon emissions from both sources was reduced from 21% to 8% and from 8% to 5%, respectively. In this study, the carbon emission factor for solid waste was assumed to be zero. However, in reality, there may be carbon emissions associated with simple processing and transportation of solid waste. Previous studies indicated that transportation distance can impact carbon emissions, and if the distance is too great, the use of solid waste may not achieve energy-saving and emission-reduction goals. Utilizing a significant amount of auxiliary cementitious material, such as fly ash, in developing environmentally friendly concrete mixtures (fly ash/cement = 3) resulted in a reduction of 411.05 t per unit. Gu et al. conducted indoor experiments with a steel slag–soil–lime mixture for roadway subgrade and found that an optimal dosage of 50% steel slag led to a reduction of 87.93 t per basic unit [48]. The calculations showed that

reusing solid waste in the material production stage could reduce carbon emissions by 17.29% per basic unit. However, there was a distinct difference between these results and other studies that proposed a 35% reduction in GHG emissions when reusing solid waste in the same basic unit. This disparity can be attributed to variations in the amount of solid waste considered. This study takes a more conservative approach in estimating carbon emission reductions from materials. Considering the diversity of data sources and different system boundaries, the results of LCA for roadway pavement comparisons remain uncertain. Nonetheless, LCA research makes a significant contribution as an auxiliary tool for evaluating and analyzing the sustainable development of the roadway industry from a forward-looking perspective.

It is noted that different carbon emission factors would yield different calculation results. Real soil remediation practices may involve the combination of multiple methods, and carbon emissions under such circumstances warrant further studies. In addition, the carbon emissions estimated using the method proposed in this study need to be validated by measured values in practice.

## 5. Conclusions

This study utilized a cradle-to-gate analysis and established a method for calculating carbon emissions from roadway construction at contaminated sites. The carbon emissions at different stages in three assumed scenarios were compared. Based on the results, the following conclusions can be drawn:

(1) Among the three described scenarios, Scenario 3 had the lowest carbon emissions of 3671.154 t per basic unit, followed by Scenario 2 with 4882.290 t per basic unit, and Scenario 1 generated the highest carbon emissions of 6050.774 t per basic unit. Thermal desorption and S/S methods reduced overall carbon emissions by 19.32% and 39.33%, respectively, compared to the traditional cement kiln co-disposal method.

(2) The analysis of carbon emissions from the three different remediation methods for contaminated soil indicate that use of the traditional cement kiln co-disposal method should be avoided or reduced due to its high carbon emissions. In thermal desorption remediation, primary remediation and tail gas treatment contributed to approximately 15% of carbon emissions, playing a significant role in the overall emissions of the remediation process. For S/S, the carbon emissions primarily came from the production of solidifiers/stabilizers, accounting for nearly 10% of the total emissions. To reduce these emissions, measures such as using eco-friendly, low-carbon raw materials and controlling transportation distances can be implemented.

(3) In roadway construction, material production was a significant contributor to carbon emissions, constituting over 45% and up to 76.82% of the total emissions. Among the materials used, asphalt had the highest proportion of carbon emissions, exceeding two-thirds of the total emissions in material production, followed by cement concrete. To mitigate these emissions, one effective approach is the reuse of solid waste, which reduced carbon emissions by 498.98 t per basic unit. By recycling and reusing solid waste materials, the need for new production of materials and their associated carbon emissions can be minimized.

**Author Contributions:** Conceptualization, L.Z.; methodology, Y.L.; validation, L.Z. and C.Q.; formal analysis, Y.L.; data collection: L.Z. and C.Q.; data curation, C.Q.; visualization, L.Z. and C.Q.; writing—original draft preparation, L.Z.; writing—review and editing, Y.D.; supervision, Y.D.; project administration, Y.D. All authors have read and agreed to the published version of the manuscript.

**Funding:** This research is funded by the National Key R&D Program of China (Grant No. 2019YFC1806000), the National Natural Science Foundation of China (Grant No. 42177133), and the Primary Research & Development Plan of Jiangsu Province (Grant No. BE2022830).

**Institutional Review Board Statement:** Not applicable.

**Informed Consent Statement:** Not applicable.

**Data Availability Statement:** Not applicable.

**Acknowledgments:** The authors wish to thank the reviewers for their time and providing us with constructive feedback. We are also would like to express gratitude for the guidance provided by Jingmin Xu and Xuening Liu.

**Conflicts of Interest:** The authors declare no conflict of interest.

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
