# Peer review of "Carbon Emission Evaluation of Roadway Construction at Contaminated Sites Based on Life Cycle Assessment Method"

_sustainability, doi:10.3390/su151612642_

Round 1

Reviewer 1 Report

General remarks:

The authors chose an interesting topic. They found an existing research gap in the application of LCA in calculating carbon emissions after the remediation and reuse of contaminated soil at road constructions. The authors soundly applied the LCA calculations on the selected field. The results are clearly demonstrated and explained. The conclusions are well supported by the results. The paper’s structure and internal balance are appropriate. 

Specific remarks:

Title: please check „ContaminatedSite” and the whole title (hard to read)

Line 13: typo („infrustructure”), please fix

Line 14: „This study aimed to carbon emission calculation…” please check the grammar

Line 17: typo („contanmiated”), please fix

Line 36: please cite the Paris Agreement

Line 220: please check the meaning of Figure 2’s title

The language is quite good, however, a cross-check of the whole text was more than welcome (see specific remarks).

Author Response

The authors would like to thank the Editor and the Reviewers for their constructive comments and suggestions. The comments are all valuable and helpful for improving our revised manuscript. All of the review comments have been carefully considered and incorporated into the revised manuscript. Itemized responses to the reviewers’ comments and suggestions are provided below.

Please note that the reviewers’ comments are shown in normal font and our responses are shown in italic fonts below. In the revised manuscript, the revised sentences/sections are marked in red. In addition to the Editor’s and the Reviewers’ comments and suggestions, we also made some other revisions to improve the manuscript. These revisions do not influence the content and framework of the manuscript; hence we do not list the changes here, but marked in red in the revised manuscript.

Reviewer 2 Report

the manuscript has several grammatical errors and typos starting with the title

I recommend 

1) better emphasise the novelty of the research in the introductory part of the text

2) better emphasise the limitations and possible future research steps in the concluding part

3) it is advisable to include a graph or flow chart that better explains the steps of the research conducted

4) more commentary accompanying all tables is necessary . At the same time, it is advisable to check that the formatting of the tables agrees with that of the journal.

5) It is advisable to include notes on the evolution of road pavements and possible applications, as well as recent bibliographical references not only from China but also from Europe and America. 

Moderate editing of English language required

Author Response

(The authors gave the same response as above.)

Round 2

Reviewer 2 Report

The manuscript still has typos and grammatical errors. It is necessary to point out in which countries the pavements described are standard and whether they refer to rigid, flexible or mixed with the appropriate purposes (roads, airports, etc.). 

once this has been corrected, the paper will be eligible for publication 

Moderate editing of English language required

Author Response

Thank you for pointing these out, we have modified and checked the whole paper. With the help of a colleague fluent in English writing, we have made revisions according to the suggested changes, including English grammar proofreading and polishing. We have added the content in line 200 and line 203 according to reviewer’s suggestion.  We have explained that roads are used on highways in line 198.
